# SAMPO: Unsupervised Knowledge Base Construction for Opinions and Implications

**Nikita Bhutani**  NIKITA@MEGAGON.AI
**Aaron Traylor**  AARON_TRAYLOR@BROWN.EDU
**Chen Chen**  CHEN@MEGAGON.AI
**Xiaolan Wang**  XIAOLAN@MEGAGON.AI
**Behzad Golshan**  BEHZAD@MEGAGON.AI
**Wang-Chiew Tan**  WANGCHIEW@MEGAGON.AI

## Abstract

Knowledge bases (KBs) have long been the backbone of many real-world applications and services. There are many KB construction (KBC) methods that can extract factual information, where relationships between entities are explicitly stated in text. However, they cannot model implications between opinions which are abundant in user-generated text such as reviews and often have to be mined. Our goal is to develop a technique to build KBs that can capture both opinions and their implications. Since it can be expensive to obtain training data to learn to extract implications for each new domain of reviews, we propose an unsupervised KBC system, SAMPO, that is based on matrix factorization techniques. Specifically, SAMPO is tailored to build KBs for domains where many reviews on the same domain are available. We generate KBs for 20 different domains using SAMPO and manually evaluate KBs for 6 domains. Our experiments show that KBs generated using SAMPO capture information otherwise missed by other KBC methods. Specifically, we show that our KBs can provide additional training data to fine-tune language models that are used for downstream tasks such as review comprehension.

## 1. Introduction

Many applications and services today are powered by knowledge bases (KBs). For instance, Google Knowledge Graph [Dong et al., 2014] supports web-search and question-answering technologies. Amazon Product Graph [Dong, 2018] support product search. Consequently, many knowledge base construction (KBC) methods have been developed in recent years [Weikum et al., 2016, Mitchell et al., 2018, Lockard et al., 2018] that focus on extracting encyclopedic knowledge in the form of real-world entities and relations from text. To the best of our knowledge, none of the KBC methods can identify implication relationships between opinions, which are abundant in user-generated text such as reviews. Reviews can express opinions about an object (e.g., `"writing"`) directly (e.g., `"pretty good writing"`), or indirectly (e.g., `"fast paced action"`). While opinions can still be identified using opinion mining, the implication relationships between opinions are missed.

Consider the first review in Example 1. It expresses an opinion about `"characters"`. General-purpose KBs [Bollacker et al., 2008, Liu and Singh, 2004] can identify that `"complex"` is linguistically related to `"complicated"` and that it can describe `"brains"`. However, they fail to identify that `"complex characters"` (from the first review) implies `"good writing"` (from the second review). This relation is (weakly) implied by the fact that both reviews describe the same movie instead of being explicitly stated in the text.

**Example.** Review$_1$: The movie has everything: fast paced action, romance and complex characters.

Opinions: (`"fast paced"`,`"action"`), (`"complex"`,`"characters"`)

Review$_2$: There's some pretty good writing going on.

Opinion: (`"pretty good"`,`"writing"`)

Understanding opinions and their implications is important to improve machine understanding of reviews for a domain. In this work, our goal is to develop KBC techniques to mine such knowledge from domain-specific reviews. For instance, we want to extract information that `"complex characters"` implies `"good writing"` from movie reviews. Such opinion KBs are useful for many downstream tasks including sentiment analysis, retrieval and question answering. For example, understanding the query "*Show me a movie with good writing*" requires the knowledge that complex characters, fast paced action imply good writing. Instead of searching reviews with `"good writing"` as a keyword, a system could expand the search using our KB to additionally look for complex characters and fast paced action. They can also provide training data to fine-tune language models and obtain a deeper understanding of reviews.

There are a number of challenges in building opinion KBs. Existing KBC methods are extractive and target knowledge that is explicitly mentioned in text. However, implications between opinions are typically not expressed explicitly in the text. Furthermore, most of the methods rely semi-supervised or supervised learning models. Obtaining human-annotated examples to train these models is expensive, especially for each domain of interest. We, therefore, aim to mine implicit relationships between opinions from a massive unlabeled review corpora. Being unsupervised, such a method can learn implicit relationships from scratch and generate necessary training data to use semi-supervised and continuous learning methods in later stages of knowledge base construction.

In this paper, we propose an unsupervised KBC system, SAMPO, for building KBs on implications between opinions from reviews. SAMPO is based on matrix factorization techniques [Koren et al., 2009], wherein each row corresponds to an item being reviewed and each column corresponds to an opinion extracted from the reviews. Since opinions and implications often do not co-occur within the same review, matrix factorization helps summarize signals across multiple reviews. Furthermore, it is unsupervised and can be applied to any domain with abundant reviews. SAMPO, thus, uses matrix factorization to learn low-dimensional representations for opinions and discover implications between opinions based on the cosine similarity of their corresponding representations. We successfully build KBs for 20 different domains using SAMPO and release them for further research[1]. We manually evaluate KBs for 6 domains and show that KBs generated using SAMPO can accurately capture implications otherwise missed by other KBC methods. We also show how KBs built using SAMPO can augment language models and benefit reading comprehension.

In summary, we make the following contributions:

- We present SAMPO an unsupervised KBC system, for capturing implications between opinions.
- We empirically show that SAMPO is domain-independent by generating KBs for 20 different domains. We manually assess the quality of 6 of the generated KBs. Our experiments show that SAMPO can consistently achieve higher precision (avg. 18%, max. 45%) and recall (avg. 6.6%, max. 16.6%) than other KBC methods.
- We publicly release the created KBs to facilitate research on review comprehension systems.

---

[1]. https://github.com/sampoauthors/Sampo

## 2. Preliminaries

We briefly introduce some notation before presenting our system. Let $\mathcal{R}$ denote a corpus of reviews about items $\mathcal{I}$ in a domain (e.g. movies). and $\mathcal{R}_i$ denote the reviews about item $i \in \mathcal{I}$. We assume that an opinion extractor $\mathcal{E}$ can extract a list of opinions $\mathcal{O}_i$ about item $i$ (e.g. *"The Dark Knight"*) from $\mathcal{R}_i$. An opinion $o \in \mathcal{O}_i$ is of the format (*modifier*, *aspect*), where *aspect* denotes the target of the opinion and *modifier* further describes the aspect. For example, $\mathcal{E}$ can extract (`"pretty good"`, `"writing"`) from the review *"There's some pretty good writing going on."*. Let $\mathcal{O}$ denote opinions about all the items in the domain.

Given a set $\mathcal{O}$ of opinions, we are interested in identifying implication relationships between opinions in the set $\mathcal{O}$. In other words, our goal is to create a knowledge base $\mathcal{K}$ of opinions and their implications. Since we are only interested in identifying implication relations, we can view opinions as nodes and implications as directed edges between the nodes. A directed edge from a head opinion $o_h$ to a tail opinion $o_t$ in $\mathcal{K}$ (denoted as $o_h \rightarrow o_t$) indicates that $o_t$ is implied by $o_h$.

## 3. SAMPO

In this section, we describe our unsupervised KBC method, SAMPO. Figure 1 illustrates SAMPO's KBC pipeline. Given a corpus of reviews and an opinion extractor, SAMPO starts by extracting all opinions from the corpus using the extractor, followed by normalization to drop slight nuances in how opinions are expressed. Once the opinions are extracted and normalized, SAMPO organizes this data into a matrix that records how often an opinion is expressed for each particular reviewed item. SAMPO then proceeds to compute a low dimensional representation for each opinion in the set $\mathcal{O}$ using matrix factorization techniques. Finally, the similarity of the learned embeddings is used to infer implications between opinions.

### 3.1 Opinion extraction & normalization

SAMPO relies on an opinion extractor $\mathcal{E}$ which can extract (*modifier*, *aspect*) pairs from a given review. Many opinion extractors have been developed recently [Samha et al., 2014, Angelidis and Lapata, 2018, Li and Lam, 2017]. Most of them have to be fine-tuned for each new domain, which can be expensive. In the absence of an extractor for new domains, a set of syntactic rules [Abbasi Moghaddam, 2013] can instead be used to extract opinions. Our approach to mine implications, however, can be used with both pre-trained extractors or rule-based extractors.

In our implementation, we use the state-of-the-art extractor developed by Li et al. [2019b] for two domains (*Hotels, Restaurants)*. Additionally, we provide a rule-based, domain-independent opinion extractor, which we refer to as MINI-MINER. It uses Spacy[2] to obtain the dependency parse trees of review sentences, and then uses a set of syntactic rules [Abbasi Moghaddam, 2013] to extract opinions. The set of rules and our implementation of MINI-MINER are publicly available[3]. Compared to state-of-the-art opinion extractors, MINI-MINER achieves a lower recall as it fails to extract opinions that are expressed in complex linguistic forms. However, it can still mine a large number of opinions with no supervision.

**Normalization:** Once all opinions are extracted, we normalize them to group *modifiers* and *aspects* that are simply minor surface-form variations. This strengthens the signals for opinions expressed

---

2. https://spacy.io/
3. https://github.com/sampoauthors/Sampo/blob/master/sampo/miniminer.py

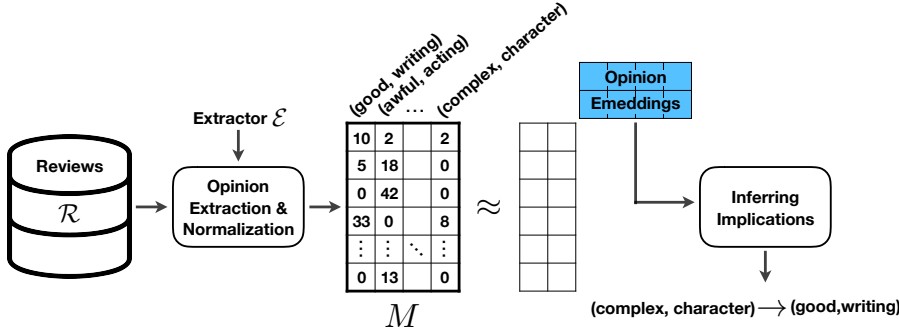

Figure 1: SAMPO's KB construction pipeline (for the movie domain).

in the corpus and improves the embeddings learned for the opinions from these signals. We normalize the aspects by dropping any *determiners* and *adverbial* modifiers, followed by lemmatization. For example, `"scrolling credits"` is normalized to `"scroll credit"`. The modifiers are normalized in the same manner, except we drop any *prepositional* modifiers. For example, `"too long"` is normalized to `"long"`. The output of this step is a list of opinions $\mathcal{O}_i$ for each item $i \in \mathcal{I}$.

### 3.2 Matrix factorization

SAMPO uses a matrix factorization technique to identify related opinions. There are two concrete reasons for using this approach: (1) Since opinions and their implications often do not co-occur within the same review, it requires techniques that summarize the signals across multiple reviews, and (2) it is unsupervised and can be applied to any domain without any overhead.

Matrix factorization enables SAMPO to learn a low-dimensional representation for each opinion which as we will show can help identify implication relations. However, there are inherent challenges when applying these techniques on extremely sparse datasets where signals from multiple reviews about an item are weak. We, therefore, target items from $\mathcal{I}$ that are frequently reviewed and opinions from $\mathcal{O}$ that occur frequently. Given the subsets of $\mathcal{I}$ and $\mathcal{O}$, we create a matrix that summarizes the frequency of each opinion across each item. Specifically, we create a matrix $M$ where each row $i$ corresponds to a unique item in the corpus and each column $j$ corresponds to a unique expressed opinion. The entry $M_{ij}$ denotes the number of times that opinion $j$ has been expressed by users when reviewing item $i$. Figure 1 shows an example of such a matrix for the movie domain.

There are notable examples in relation extraction that use factorization techniques over binary matrices [Riedel et al., 2013]. We could adopt a similar approach and construct $M$ such that $M_{ij}$ is 1 if opinion $j$ is expressed at least once for item $i$ and 0 otherwise. While this approach works well for a factual KB where a candidate fact is either true or false, it is not suited for mining implications between opinions from reviews. Reviews can have conflicting opinions about the same item although with significantly different frequencies. In order to capture these signals, SAMPO uses frequencies of opinions to construct the matrix $M$.

Once the matrix $M$ is created, SAMPO computes a $d$-dimensional embedding for each item $i$ (denoted as $\mathbf{v}_i$) and for each opinion $j$ (denoted as $\mathbf{v}_j$). We provide guidelines for tuning hyperparameter $d$ based on stability and error metrics, which we will introduce shortly. We empirically found that the results were not sensitive to $d$ in the range 20-50.

In general, low-dimensional embeddings are considered effective if they can reconstruct the original dataset. In our case, we train embeddings such that $M_{ij}$ values can be derived from the embeddings $\mathbf{v}_i$ and $\mathbf{v}_j$. There are several methods for approximating the $M_{ij}$ values from embeddings. Following the rich literature on matrix factorization [Lee and Seung, 2001], SAMPO computes embeddings such that their inner product which we denote as $\widehat{M}_{ij} = \mathbf{v}_i \mathbf{v}_j$ would be a good approximation of $M_{ij}$. Formally, SAMPO computes embedding to minimize the following error value which we refer to as the *reconstruction error*: $E_{rec} = ||M - \widehat{M}||_F$.

### 3.3 Inferring Implications

Given the embeddings learned in the previous step, SAMPO identifies implications between opinions by constructing a nearest neighbor graph based on the similarities between the embeddings. Specifically, for each opinion $o_h$ in $\mathcal{O}$, SAMPO finds the top $k$ candidate opinions for $o_t$ based on their similarity. We refer to these opinions as the neighbors of opinion $o_h$ which we denote as $N_k(o_h)$, and we create an implication edge $o_h \to o_t$ for each neighbor $o_t \in N_k(o_h)$.

To construct this nearest neighbor graph, we need to select a metric for measuring the similarity of opinion embeddings. Euclidean distance, inner product of embedding vectors, and cosine similarity are natural candidates to consider for this task. We argue that Euclidean distance and inner product are not suitable for this setting. Different opinions have different frequencies in the corpus. The embedding of frequent opinions such as ("good", "movie") have larger $l_2$ norm values so that the vectors can reconstruct the high-values associated with the column in matrix $M$. Similarly, infrequent opinions have smaller $l_2$ norms. Both Euclidean distance and inner products are sensitive to the norm of vectors. They fail to capture the similarity between opinions that have a similar trend of values in matrix $M$ but with different magnitudes. To avoid the need for another normalization, SAMPO, therefore, uses cosine similarity.

While the similarity scores between opinions are symmetric, the $k$-nearest neighbor graph constructed is a *directed* graph. This enables SAMPO to identify implication relations. For instance in the hospitality domain, we observe that `"bad breakfast"` is among the top 5 neighbors of `"burnt coffee taste"`, but `"burnt coffee taste"` does not appear in the top 5 neighbors of `"bad breakfast"` and more generic opinions such as `"limited options"` or `"poor service"` are closer to `"bad breakfast"`. This asymmetry implies that there should be a directed edge from `"burnt coffee taste"` to `"bad breakfast"` and not vice versa. On the other hand, `"poor breakfast"` and `"bad breakfast"` appear in the top 5 nearest neighbors of each other and we can conclude that these opinions imply each other and are likely to be equivalent.

While cosine similarity proves to be a better choice for constructing the neighborhood graph, it's not a stable metric for infrequent opinions. In other words, small changes in the frequencies of an infrequent opinion in matrix $M$ can have a large effect on its similarity scores. We discuss how SAMPO addresses this instability problem next.

**Handling Instability:** To ensure that high similarity scores are not due to random co-occurrences of infrequent opinions, SAMPO repeats the factorization process on noisy versions of the matrix $M$ for a few iterations (i.e., 5 in our setting). Adding noise would change the embedding vector which in turn would affect the similarity scores. SAMPO, thus, uses the average of similarity scores. The success of this technique highly depends on how much noise we add to the matrix and how we model this noise. We refer to the theoretical insights about noise modeling in matrix factorization literature and assume that matrix $M$ can be approximated with a low-rank matrix $\widehat{M}$ and that the

| Source | Dataset/Domain | # Reviews | # Items | # Opinions | Example Opinion |
|--------|----------------|-----------|---------|------------|-----------------|
| Amazon | Movies | 1.69M | 49K | 2.09M | (intricate, plot) |
| | Electronics | 1.68M | 61K | 1.87M | (excellent, sound) |
| | Books | 8.89M | 360K | 8.59M | (complex, plot) |
| | CD Vinyl | 1.09M | 63K | 1.45M | (good, lyric) |
| TripAdvisor | Hotels | 18k | 208 | 134K | (friendly, staff) |
| Yelp | Restaurants | 176k | 860 | 1.41M | (romantic, dinner) |

Table 1: No. of reviews, items and opinions in each domain.

error matrix $\Sigma = M - \widehat{M}$ follows a Gaussian distribution [Chi and Kolda, 2010]. This implies that we can create a noisy version of matrix $M$ by (1) factorizing matrix $M$, (2) estimating the parameters of the Gaussian distribution from the error matrix $\Sigma$, (3) sampling noise values from the estimated Gaussian distribution and adding them to values in matrix $\widehat{M}$.

We also conduct experiments with a version of SAMPO that does not model or add any noise which we call SAMPO$_{Basic}$. However, since the tools we use for factorization include stochastic elements in their initialization and optimization procedures, we still repeat the factorization for a few iterations and aggregate the similarity scores to produce robust and stable results.

## 4. Experiments

Given reviews from a domain, can SAMPO accurately build an opinion KB, and can other KBC methods capture such information? In this section, we seek answer to this question empirically. We target multiple domains and build KBs using SAMPO and other KBC methods. We assess each KB by manually examining sample implication relationships. Our experiments show that KBs built using SAMPO achieve higher average precision (avg. 18%, max. 45%) and recall (avg. 6.6%, max. 16.6%) than other approaches. We also show that our KBs capture information otherwise missed by pre-trained language models, and hence can complement language models in downstream tasks.

### 4.1 Datasets and experimental setup

**Datasets:** We built opinion KBs by running SAMPO on reviews from over 18 product categories in Amazon [He and McAuley, 2016], hotel reviews from TripAdvisor [Marcheggiani et al., 2014] and restaurant reviews from Yelp[4]. Since evaluating the quality of all constructed KBs is expensive, we focus on evaluating the KBs built for the 4 largest Amazon categories, hotels and restaurants.[5]. Table 1 shows the statistics of these datasets. For each dataset, we construct a test set by selecting a sample of 200 opinions (*query*). We use frequency of opinions as sample weights to ensure that the test set is a good representation of opinions in the dataset.

**Setup:** We use Tensorly's implementation of PARAFAC [Kossaifi et al., 2019] for factorization and aggregate the results across 5 iterations. We report *instability* in terms of average standard deviation in cosine similarity scores across iterations [Antoniak and Mimno, 2018]. We also report average

---

4. https://www.yelp.com/dataset
5. We release other KBs at https://github.com/sampoauthors/Sampo for future research.

percent overlap of *tail* opinions for each *query* opinion across iterations [Wendlandt et al., 2018]. Note that we used pre-trained opinion extractor [Li et al., 2019a] for *Hotels* and *Restaurants* and MINI-MINER for the Amazon datasets.

**Baselines:** For a fair comparison, we compare SAMPO and its variant with no noise modeling, SAMPO_Basic against 3 unsupervised KBC methods. For each *query* opinion in the test set, each method returns a ranked list of *tail* opinions implied by the query.

- **Point-wise Mutual Information (PMI)**: PMI gives a measure of association of opinions. For opinions $o_h$ and $o_t$, we measure PMI as $PMI(o_h, o_t) = log\ p(o_h, o_t)/p(o_h)p(o_t)$. We use the frequency of opinions and co-occurrence counts to compute the PMI scores. For each *query* opinion in the test set, we rank all other opinions in $\mathcal{O}$ based on their PMI scores.
- **Universal Schema**: Given a relation schema, this approach predicts missing relations given a pair of entities. In the absence of apriori known relations in our setting, we define a unary *hasA(o)* relation from each opinion $o$ (e.g., *hasA("slow pace")*). We consider an item to have the *hasA(o)* relation if the opinion $o$ is expressed more than 5 times for the item (e.g., `"Magnolia"` *hasA("slow pace")*). We then use the universal schema method to obtain representations for each relation and use these representations to find implications in the same manner as SAMPO. This baseline is akin to factorization over binary matrices.
- **Pre-trained language model**: We used a variant of LAMA knowledge probing approach [Petroni et al., 2019] to investigate if language models can be used to generate opinion KB. Given a query opinion $o_h$, we first create a sequence "$o_h$ implies [MASK] [MASK]", where the first [MASK] denotes a modifier and the second [MASK] denotes an aspect of an implied opinion $o_t$. We use BERT's masked language model to predict a likely modifier *mod* for the first [MASK]. We then create another sequence "$o_h$ implies *mod* [MASK]" and let the model predict a likely aspect *asp* for the second [MASK]. We then use the joint probability to rank all possible candidates for $o_t$. In our experiments, we have used the *uncased large* BERT model which has 24-layer, and 340M parameters. Note that this method is only capable of dealing with opinions that consist of single-token modifier and aspects. To be fair, we report the evaluation metrics for this baseline only this relevant subset of opinions.

**Evaluation:** Each KBC method provides a ranked list of *tail* opinions implied by a query in the test set. We retain the top 5 tail opinions for each query. This helps us collect a sample of directed edges from an opinion KB built using each method. We pool the edges from each method and ask crowdworkers to manually judge their "truth". The inter-annotator agreements for this task are reported in the appendix. In the process, we discard any noisy opinions that can be attributed to the opinion extractor. This gives us a set of labeled edges to calculate precision and recall of each system. Since each system produces an ordered list of edges for each query, we compute mean average precision (MAP) for the set of query opinions. In addition, we compute overall precision and pseudo-recall for different values of top $k$ and present them across a PR curve.

**Hyper-parameters:** Each dataset differs in the distribution of opinions across items. Intuitively, this affects the sparsity of the matrix to be factorized, which can further affect the resulting KB. We tuned the number of most-reviewed items [2000,20000] and the number of most-frequent opinions [2000,5000] for large datasets from Amazon. Empirically, we found item count 2000 and opinion count 2000 yielded best results across domains. We also tuned the embedding dimension [10,20,30,40,50] for each domain based on reconstruction error and instability of factorization. We report the best dimension size in Table 2.

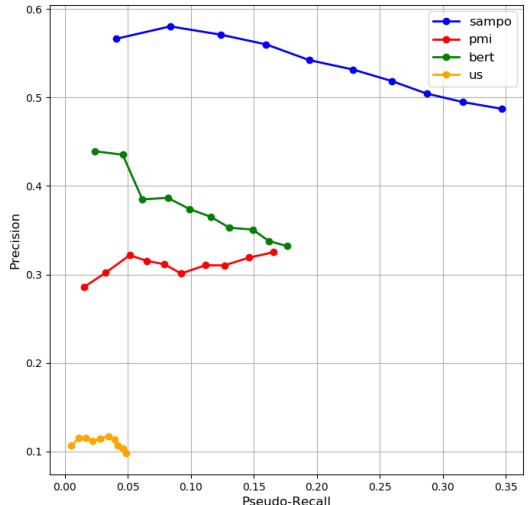

| Domain | $E_{rec}$ | $\sigma_{CS}$ | Overlap |
|---|---|---|---|
| Movies | 0.6888 | 0.0368 | 64.97 |
| Electronics | 0.6131 | 0.0258 | 75.73 |
| Books | 0.5648 | 0.0613 | 57.21 |
| CD Vinyl | 0.6655 | 0.0622 | 50.53 |
| Hotels | 0.0977 | 0.0802 | 40.09 |
| Restaurants | 0.4781 | 0.0596 | 58.65 |

Table 3: Recons. error (RE), mean std-dev of cosine similarities ($\sigma_{CS}$) and mean percent overlap of edges (Overlap) across 5 iterations.

Figure 2: PR curve of KBC methods at different values of top $k$ for Electronics domain.

## 4.2 Results and Discussion

**Effectiveness over other KBC methods.** Table 2 shows the MAP of different KBC methods on various domains when top-5 tail opinions for each query were considered. We found that SAMPO and SAMPO$_{\text{Basic}}$ achieve highest MAP scores across all the domains except *Restaurants*. Figure 2 compares the PR curves for various KBC systems in *Electronics* domain. Note that SAMPO outperforms other KBC systems across all recall levels. This shows that our approach can effectively discover implications. This finding is also consistent with the MAP scores of SAMPO. The PR curves and example predictions for other datasets are reported in the appendix.

**Effectiveness of noise modeling.** As shown in Table 2, SAMPO mostly outperforms SAMPO$_{\text{Basic}}$, indicating the benefit of noise modeling. The only significant exceptions are the *Hotels* and *Restaurants* datasets which both have a few number of reviewed items. We empirically illustrate that using SAMPO$_{\text{Basic}}$ is preferable for domains where limited data is insufficient for noise modeling.

| Dataset | Best dim | SAMPO | SAMPO$_{\text{Basic}}$ | PMI | Univ. Schema | Language Model |
|---|---|---|---|---|---|---|
| Movies | 30 | 0.4926 | **0.5047** | 0.2548 | 0.0983 | 0.3794 |
| Electronics | 30 | **0.4477** | 0.4243 | 0.2142 | 0.0609 | 0.2796 |
| Books | 40 | **0.3422** | 0.2955 | 0.1656 | 0.0919 | 0.3073 |
| CD Vinyl | 50 | **0.2692** | 0.2575 | 0.1881 | 0.1702 | 0.2506 |
| Hotels* | 30 | 0.3611 | **0.4256** | 0.0993 | 0.1750 | 0.3519 |
| Restaurants* | 30 | 0.3869 | 0.4116 | 0.1975 | 0.1233 | **0.4724** |

Table 2: MAP@5 of KBC methods. The second column shows the embedding dim used in SAMPO. * We used pre-trained opinion extractor for *Hotels* and *Restaurants*.

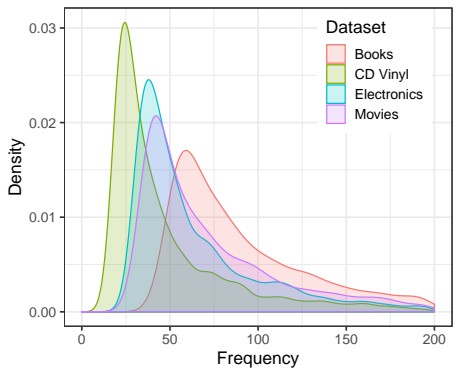 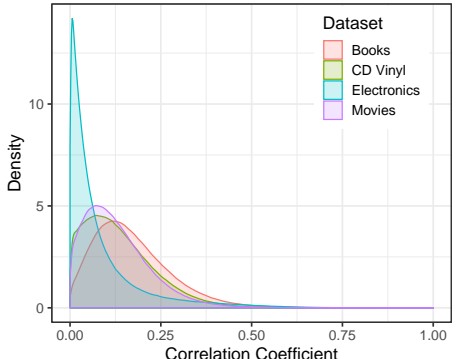

Figure 3: Distribution of opinion frequencies across different datasets

Figure 4: Correlations between the expressed opinions of reviewed items

**Stability of factorization.** Table 3 shows the reconstruction error, mean standard deviation of cosine similarities and mean percent overlap of tuples for different iterations of SAMPO . Interestingly, despite the high reconstruction error, the cosine similarity scores and rankings do not vary a lot across iterations. This indicates that we can obtain stable KB with a small number of iterations.

**Performance across datasets.** We focused on 4 datasets from Amazon of comparable sizes to investigate how dataset characteristics impact SAMPO's performance. We found two key factors that affect performance: (a) frequencies of extracted opinions and (b) diversity of their patterns across different items. Figure 3 shows the distribution of frequencies of opinion for different datasets. Many infrequent opinions (as in *CD Vinyl*) lead to weak signals, and therefore subpar performance. Figure 4 shows the distribution of similarities of all pairs of items, where similarity of a pair is the correlation of their opinions. High overlap between opinions expressed on different items (as in *Books*) makes it difficult to distinguish how opinions relate to each other. On the other hand, small degree of similarity (as in *Electronics*) makes it easy to determine which opinions are related.

**Qualitative Evaluation & Discussion.** Figure 5 shows the embeddings for example query opinions from *Electronics*. As shown, relevant opinions are clustered together in the embedding space. For instance all yellow dots in Figure 5 are positive opinions about image quality. SAMPO relies on such neighborhood structures to infer the implications between opinions. To make sure that the points clustered in these neighborhoods have a good diversity of aspect, we looked into the correct predictions of SAMPO and found that it is usually not biased towards finding tail opinions which share the same aspect as the query opinion. Specifically, we found that on an average 42.9% of the correct predictions had different aspects for query and tail opinions.

SAMPO outperforms other KBC methods across all but one domain. We, therefore, looked closely into our Language Model baseline. We found that a language model has a lower bias toward predicting the same aspect for the tail opinion. Compared to the 42.9% implications with different aspects that SAMPO mines, our language model baseline finds 52.0% such implications. Although for most datasets, this diversity of aspects comes at the cost of making more incorrect predictions. Nevertheless, this suggests that SAMPO can further enhance the quality of KBs by incorporating signals from pre-trained language models such as BERT. At the same time, our experiments in the following section, demonstrate that pre-trained language models fail to predict some of the impli-

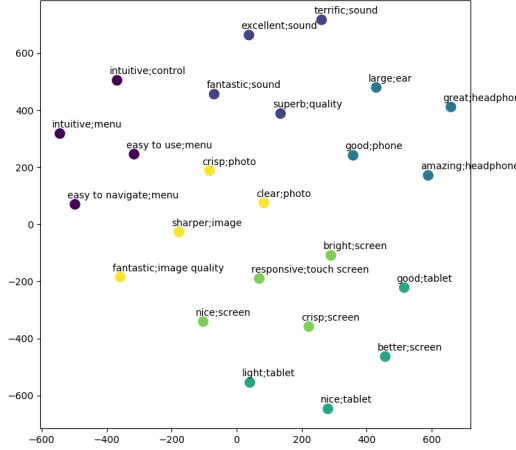

Figure 5: Embeddings for query opinions and their tail opinions from Electronics.

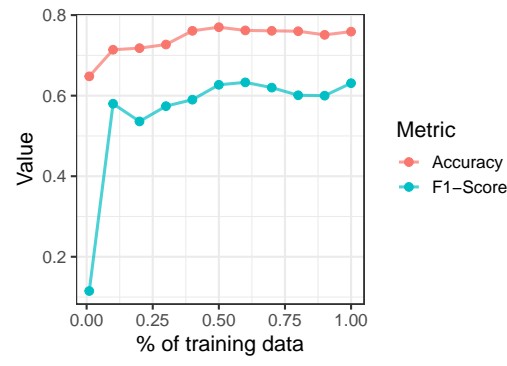

Figure 6: BERT's performance in predicting implications for Electronics in a supervised setting

cations that SAMPO discovers, even when we rely on supervised methods. We leave the problem of creating a system that combines the power these complementary methods for future work.

### 4.3 Beyond pre-trained language models

We investigate if pre-trained language models already know about the implications we mine. If these models fail to identify them, then our KBs can provide additional training data to complement these models. To avoid bias towards a chosen task and fine-tuning method, we adopt a straightforward method to demonstrate the usefulness of our KBs. Specifically, we investigate if pre-trained language models can identify the implications that SAMPO mines *in a supervised setting*. To do so, we fine-tune a pre-trained BERT model [Devlin et al., 2019] with a subset of our KB which was verified by crowdworkers as correct implications. Then, we evaluate the fine-tuned model on a held-out subset. We call this model a *knowledge verification model*. It takes as input a pair of opinions, $o_h$ and $o_t$ and predicts a label 1 if $o_h$ implies $o_t$ or 0 otherwise.

Our implementation is based on HuggingFace [Wolf et al., 2019]. For each domain, we split a labeled set of directed edges (80:20) for training and evaluation. To understand the influence of the size of training data on the model's performance, we trained multiple models with various fractions of the training data. Specifically, we used the following fractions: [0.1, 0.2, 0,3, 0.4, 0.5, 0.6, 0.7, 0.8, 0.9, 1.0]. We trained each model 20 epochs with a learning rate of 2e-5.

Figure 6 shows the performance of the knowledge verification model on the test dataset for *Electronics*. As expected, the accuracy and F1-score of the model generally improve when the size of the training data increases, but plateau after 50% of training data. The accuracy of the verification model peaks to about 80%, indicating that even supervised language models fail to predict 20% of the implication relations. This suggests that the opinion KB built using SAMPO contains implications that can not be trivially obtained by fine-tuning language models. In fact, the performance of knowledge verification models using the entire training data from other 5 datasets could not improve the accuracy over 80% with the lowest accuracy (i.e., 0.65%) on *Movies* and highest accuracy (i.e., 77%) on *Hotels*.

## 5. Related Work

**Knowledge base construction:** Prior work on KBC uses expert knowledge [Suchanek et al., 2007, Bollacker et al., 2008, Liu and Singh, 2004, Auer et al., 2007] and models entities and relations from encyclopedic text [Nakashole et al., 2011, Carlson et al., 2010]. These methods often require labeled training data to learn to map mentions to pre-defined entities and relations. We focus on modeling opinions and their implications, which are highly domain-specific. Modeling them, thus, requires an open schema. Open IE [Etzioni et al., 2008], is an alternative approach to build KBs without pre-defined entities and relations but it can only extract explicitly mentioned facts. In contrast, SAMPO finds opinions and their implications which are typically unstated in the text.

**Relation extraction via factorization:** Using tensor factorization techniques has been studied extensively for relation extraction [Riedel et al., 2013, Chang et al., 2014, Nimishakavi et al., 2016, Rocktäschel et al., 2015]. These techniques focus on predicting missing relations between entities and assume the relation schemas are given. Our setting, however, is fundamentally different. SAMPO does not assume access to an incomplete KB or relation schema. It starts with no prior knowledge about opinions and how they are related. The setting in this paper is unsupervised.

**Opinion mining from reviews**: There is large body of work in opinion mining and sentiment analysis to extract aspects from reviews, in addition to their associated opinion and orientation [Poria et al., 2016, Hamilton et al., 2016, Wang et al., 2016, Hu and Liu, 2004]. The extraction task is known as aspect term extraction [Brody and Elhadad, 2010, He et al., 2017, Angelidis and Lapata, 2018]. Recent works [Li et al., 2019a] are based on BERT pre-trained model [Devlin et al., 2019] and have been shown to achieve state-of-the-art performance. Recent progress in sentiment analysis [Pontiki et al., 2016] has enabled systems decide whether a particular review text about an aspect of a product/service is positive or negative. SAMPO provides a framework to incorporate such systems and build higher quality KBs for review comprehension.

## 6. Conclusion

We presented SAMPO, an unsupervised method for building KBs of opinions and their implications from review corpora. We discussed that discovering implications of expressed opinions requires methods which can consider multiple reviews simultaneously. This makes matrix factorization a suitable tool for SAMPO. Through a range of experiments, we showed that SAMPO outperforms various unsupervised KB construction baselines and achieves higher precision (avg. 18%) and higher recall (avg. 6.6%). Finally, we released SAMPO as well as the constructed KBs on 20 domains to facilitate future research and industry applications.

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

# Appendix A. Precision-Recall Curves

Figures 1-6 provide the precision-recall curve for all other domains missing from the paper.

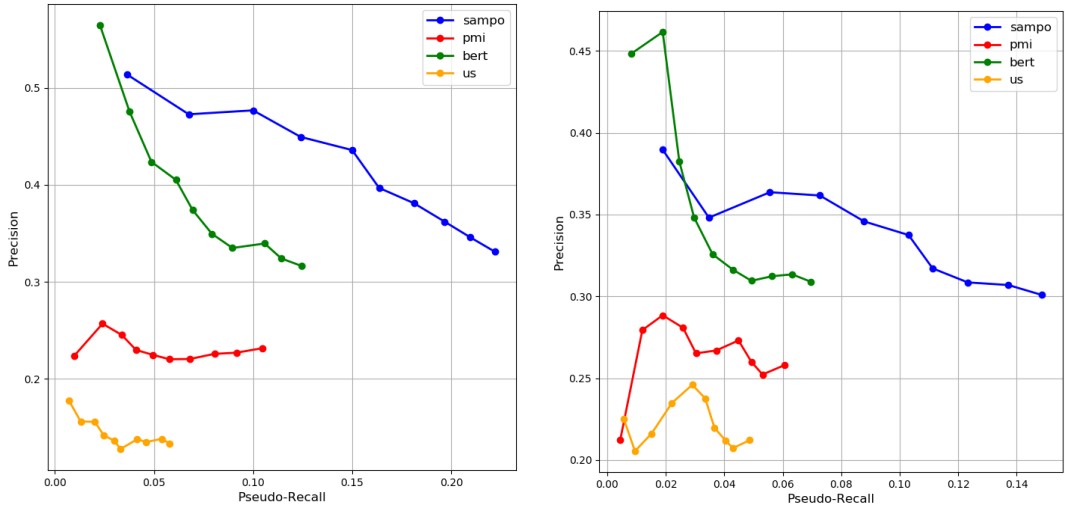

Figure 7: Precision-Recall curve at different values of top $k$ for Books domain

Figure 8: Precision-Recall curve at different values of top $k$ for CD Vinyl domain

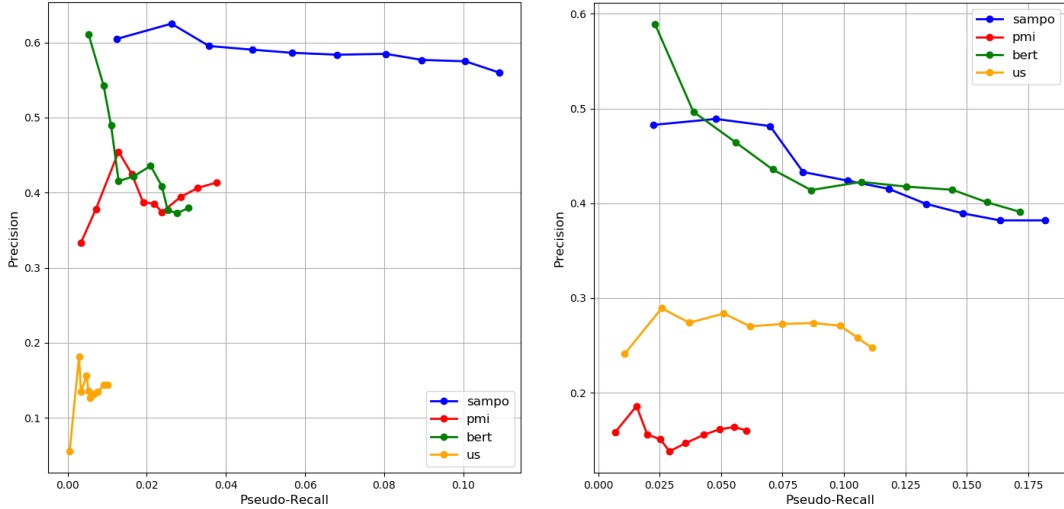

Figure 9: Precision-Recall curve at different values of top $k$ for Movies domain

Figure 10: Precision-Recall curve at different values of top $k$ for Hotels domain

# Appendix B. Inter-Annotator Agreements

To minimize the cost of evaluating our framework, we have collected a single judgement on each implication mined by SAMPO. However, to ensure that we observe a reasonable degree of agreement between annotators, we have randomly selected 70 test questions (i.e, 10-15 per dataset) and asked

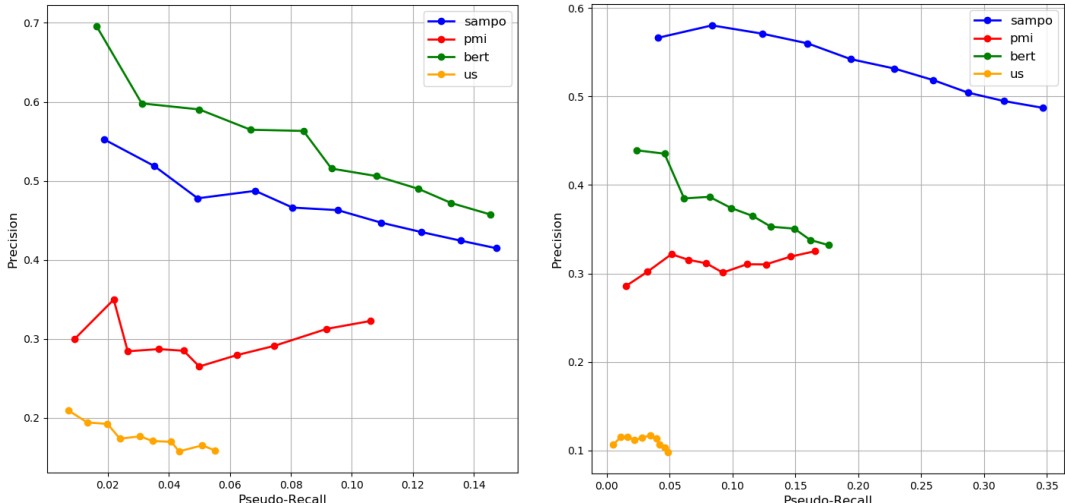

Figure 11: Precision-Recall curve at different values of top $k$ for Restaurants domain

Figure 12: Precision-Recall curve at different values of top $k$ for Electronics domain

all annotators (i.e., about 1200) to annotated all test questions. Based on the collected data, we have observed an average *Fleiss' Kappa* agreement score of 0.75 when considering the following four categories: $\{o_1 \rightarrow o_2, o_2 \rightarrow o_1, o_1 \leftrightarrow o_2, \text{ and } o_1 \not\leftrightarrow o_2\}$. The agreement score per each dataset are listed in Table 4.

| Source | Dataset/Domain | # Test Questions | # Annotators | # Fleiss' Kappa |
|---|---|---|---|---|
| Amazon | Movies | 15 | 2107 | 0.71 |
| | Electronics | 14 | 1274 | 0.75 |
| | Books | 10 | 1895 | 0.72 |
| | CD Vinyl | 11 | 1726 | 0.73 |
| TripAdvisor | Hotels | 10 | 1579 | 0.78 |
| Yelp | Restaurants | 10 | 1628 | 0.78 |

Table 4: The inter-annotator agreements in each domain.

# Appendix C. Examples of Mined Implications

In this section, we simply provide a sample of correct and incorrect implications that SAMPO has mined in each domain. These examples are listed in Table 5.

| Source | Domain | Correct | Head Opinion | Tail Opinion |
|---|---|---|---|---|
| Amazon | Movies | ✓ | (satisfying, ending) | (good, script) |
| | | ✓ | (believable, acting) | (interesting, movie) |
| | | ✓ | (great, art direction) | (great, production design) |
| | | ✓ | (amateurish, acting) | (not that great, acting) |
| | | ✗ | (great, soundtrack) | (great, scene) |
| | | ✗ | (brilliant, music) | (complex, film) |
| | Electronics | ✓ | (responsive, key) | (nice, keyboard) |
| | | ✓ | (bright, picture) | (great, tv) |
| | | ✓ | (easy to navigate, menu) | (simple, interface) |
| | | ✓ | (tinny, sound) | (aweful, quality) |
| | | ✗ | (awesome, sound) | (good, customer support) |
| | | ✗ | (cheap, unit) | (large, size) |
| | Books | ✓ | (graphic, violence) | (disturbing, book) |
| | | ✓ | (interesting, twist) | (unpredictable, story) |
| | | ✓ | (unrealistic, ending) | (horrible, ending) |
| | | ✓ | (difficult to read, book) | (hard to read, book) |
| | | ✗ | (predictable, story) | (fun, story) |
| | | ✗ | (clean, writing) | (intriguing, story) |
| | CD Vinyl | ✓ | (good, melody) | (powerful, song) |
| | | ✓ | (awesome, lyrics) | (powerful, album) |
| | | ✓ | (phenomenal, album) | (simple, song) |
| | | ✓ | (powerful, voice) | (outstanding, vocals) |
| | | ✗ | (familiar, song) | (silly, song) |
| | | ✗ | (fast, song) | (heavy, album) |
| TripAdvisor | Hotels | ✓ | (excellent, service) | (amazing, hotel) |
| | | ✓ | (good, drinks) | (great, bar) |
| | | ✓ | (clean, room) | (good, experience) |
| | | ✓ | (friendly, staff member) | (helpful, employees) |
| | | ✗ | (quiet, rooms) | (efficient, staff) |
| | | ✗ | (nice, room) | (great, city view) |
| Yelp | Restaurants | ✓ | (great, meal) | (good, restaurant) |
| | | ✓ | (fast and friendly, service) | (quick, food) |
| | | ✓ | (delicious, menu) | (tasty, dishes) |
| | | ✓ | (fresh, fish) | (good, sushi) |
| | | ✗ | (good, nachos) | (great, music) |
| | | ✗ | (knowledgeable, staff) | (classy, atmosphere) |

Table 5: Examples of correct/incorrect mined implications.

