# OpenReview forum: "Sampo: Unsupervised Knowledge Base Construction for Opinions and Implications"
_AKBC.ws/2020/Conference — AKBC 2020_

### Official Review · AnonReviewer2 · 2020-03-23
**Clearly written and motivated paper, but need more analysis and explanation for experiments**

**Rating:** 6
**Confidence:** 4

**Review:**

This paper addresses the problem of building KBs from datasets like product reviews, that identifies implications between opinions from reviews. The authors claim that opinions and their implications often do not co-occur within the same review and the annotation for such implication relation is expensive, for which matrix factorization techniques turn to be a promising approach.

Strength:
1. This paper is well written and most contents are easy to follow. The motivation is clear and the method description is well presented with reasoning behind each component choice.
2. The authors released the results of generated KB from their model on 20 domains, which could be useful for the research community.
3. Their experiment results seem to be in favor of their proposed method, although it's a very simple method.

Weakness and Questions:
1. Intuitively, the "implication" relationship proposed in this paper should be directed. A implies B doesn't mean B implies A. However, the method introduced in the paper decides such implication relationship based on cosine similarity, which is symmetrical.
2. Please provide more details about how you obtain representations from universal schema, as this seems to be the major season why you have the huge performance gap between your model and universal schema.
3. For using pre-trained LMs, which BERT model did you use?
4. When applying pre-trained models, why do you mask out the aspect token when predicting the mod token?
5. Do you restrict your modifier and aspect tokens to be unigrams? If yes, you should clarify this in the paper, as this is inconsistent with motivation examples you provided. If not, this is not comparable between pre-trained models and other models. And this is very important as the difference between your model and pre-trained LM doesn't seem to be big.
6. In section 4.3, if you are feeding LM supervision from generated KB, this is technically not "supervised" as there are quite a lot of noise from the generated KB. So it's hard to tell whether the reason why such "supervised" model cannot achieve good results is due to the limitation of LM or the noise from training. Thus, your conclusion in 4.3 needs more convincing evidence.
7. Again in section 4.3, you say "It takes as input a pair of opinions, $o_h$ and $o_t$ and predicts a label 1 if $o_h$ implies $o_t$ or 0 otherwise." It seems like you don't feed the LM any context of such extracted opinions. This is not a typical setting for LM, although I understand it's hard to incorporate such "context" in your setting.
8. There is little analysis of comparison across domains.
9. More prediction examples and error analysis are needed


Some minor points in writing:
1. I found it hard to follow in the second paragraph of second page, that starts with "There are a number of challenges in ...". I didn't understand until I finished reading some later content.

2. The authors may want to provide more details about MINI-MINER. For example, whether you use any NLP tools to extract certain linguistic features? And did you use this for baselines as well?

3. Although the authors give a brief description about "factorization techniques over binary matrices" and why they think this will cause worse results, I was expecting some ablation analysis on this, while they didn't provide any.

4. In section 3.3, I think the last $o_t$ in the first paragraph should be $o_h$ instead.

5. How do you calculate the probability in PMI? Do you use frequency based counts? If yes, clarify.

6. I think the authors need better experiment figures to be put in the paper. The current ones seem very rough and some are with bad resolutions.

---

> ### Author Response · Authors · 2020-04-08
> **Response to Reviewer #3 [1/1]**
>
> R3W1: Q: How implications (which are directed) are derived from cosine similarities which are symmetric?
> A: We have updated Section 3.3 to reflect this discussion better. The proximity between the opinions embeddings identifies similarity. However, the k-nearest graph constructed using these similarities is a directed graph. The directions of edges enable Sampo to discover implications. For instance, we see that “bad breakfast” is among the top 5 neighbors of “burnt coffee taste”, but “burnt coffee taste” does not appear in the top neighbors of “bad breakfast” and more generic opinions such as “limited options”, “poor service” are closer to “bad breakfast”. This helps us identify that there should be a directed edge from “burnt coffee taste” to “bad breakfast” and not vice versa. On the other hand, “poor breakfast” and “bad breakfast” appear in the top neighbors of each other and we can conclude that the implication edge goes both ways which further implies the two opinions are equivalent.
>
> R3W2: Q: How representations are obtained from Universal Schema?
> A: We considered the items being reviewed as entities and considered each opinion O as a unary relation which we called “has-a-O”. For instance, we consider “Casablanca” has-a-(romantic,theme). Thus, our matrix has as many relations as our extracted opinions. Universal schema computes a representation for each relation and we use these representations to measure the similarity between different opinions. We have updated our description of our setup in Section 4.1.
>
> R3W3: Q: Which BERT model is used?
> A: We used the uncased large BERT mode which has 24-layers, hidden state of size 1024, 16 self-attention heads, and 340M parameters. We have included these details in the new uploaded revision.
>
> R3W4: Q: Why do you mask out the aspect token when predicting the mod token?
> A: Our goal is to identify the extent to which pre-trained models (e.g., BERT) can identify implications between opinions. As is mentioned in the paper, the baseline computes the joint probability of a MOD- ASP pair to be able to rank the likelihood of an implication relation. We compute this joint probability as
> P(MOD-ASP) = P(MOD) * P(ASP|MOD) which is why in the first step (which corresponds to the first term), the ASP is masked.
>
> R3W4: Q: Are modifiers and aspects restricted to be unigrams (in the LM baseline)?
> A: We restrict the modifiers and aspects to be unigrams only for the pre-trained LM baseline, following the same approach as the LAMA knowledge probing approach by Petroni et al. 2019. To be fair in our evaluation, we measure the precision and recall of this baseline on the subset of opinions which consist of unigram modifiers and aspects. We will clarify this further in the paper.
>
> R3W5: Q: Is the “supervised” method valid if you use the generated KB for supervision?
> A: We should clarify that to train the supervised technique, we presented our KB to crowdworkers to filter the incorrect implications. We used the labeled data to train the supervised methods. We have adjusted the discussion in Section 4.3 to reflect this.
>
> R3W6: Q: Did you consider providing the context of opinions to the LM?
> A: This is a valid point as LMs might be able to capture more given the “context”. In our setting, an opinion might appear in multiple contexts, so besides setting up the architecture to incorporate the context, we should decide which contexts should be fed to the system. This is an interesting direction to explore and thanks for pointing that out.
>
> R3W7: Q: More analysis across domains?
> A: We agree that comparing the performance across domains would be an interesting analysis to conduct. Our focus in this paper was on evaluating the quality of the KB and conducting experiments to find the best technique for mining implication relations from reviews.
>
> R3W8: Q: More prediction examples and error analysis?
> A: We have included more examples and point to the complete set of predictions in the repository.
>
> R3W9: Q: Clarify the second paragraph of the second page.
> A: Thank you for your feedback. We have rephrased the discussion.
>
> R3W10: Q: Provide more details on Mini-Miner.
> A: We have included more details on Mini-Miner in the text. We have used package spacy to obtain the dependency parse trees of review sentences and used hand-written patterns to find opinions.
>
> R3W11: Q: Explore factorization techniques over binary matrices.
> A: In our experiments, Universal Schema serves as the binary-version of our technique. We have clarified this in the experimental setup.
>
> R3W12: Q: Typo in section 3.3
> A: Thank you for pointing out the typo. We have fixed it in the revision.
>
> R3W13: Q: How PMI is calculated?
> A: Yes, we calculate the PMI based on frequency of opinions and co-occurrence counts. We follow Church et al. 1989 (https://www.aclweb.org/anthology/P89-1010/) to compute PMI. We have updated Section 4.1 to describe this more clearly.
>
> R3W14: Q: Improve the figures.
> We have improved the figures in the paper.

---

> > ### Comment · AnonReviewer2 · 2020-04-18
> > **Thanks for adding clarifications**
> >
> > Thanks for providing clarifications to some major points like implication directions, etc. I think the paper is in a better shape now.

---

### Official Review · AnonReviewer1 · 2020-03-27
**Interesting kind of KB but missing discussion of assumptions**

**Rating:** 6
**Confidence:** 4

**Review:**

The authors propose a method to automatically build a Knowledge Base of opinions and implications between them. The KB is realized as a directed graph where nodes correspond to opinions in a canonical form (modifier, aspect), and edges indicate implications. It is built by factorizing a matrix of item-opinion frequencies, and finding the top k neighbors of an opinion.

The idea of creating a KB of opinions is relevant for the field of KB construction, and it opens the door to further research where these graphs can be used.


Strengths
- The proposed method allows to obtain a KB of opinions from raw text as input. For cases where the performance of an opinion extractor can be harmed due to a change in the domain, the authors propose a set of rules.
- The authors propose a method for improving robustness, based on optimization under noisy data.
- The experiments are thorough, showing results with multiple datasets from different domains, and relevant baselines.
- Overall, the paper is clear and well structured.


Weaknesses

The paper starts with the promise of modeling implications between opinions, but later I realize that this modeling is limited by assumptions and design, that are not addressed.

The first is: implications between opinions are, as across all NLP, very sensitive to relatively small lexical variations in text. Take for example the opinions O1 = "The movie has complex characters" and O2 = "There's good writing going on". The proposed pipeline would then give as candidate implication O1 -> O2. What if O1 changes to "The movie has unnecessarily complex characters"? How does the opinion extractor behave in these cases? How does this affect performance?

The second, and probably more crucial, is: the directions of the implications are not really modeled by the proposed method. In fact, following the example above, the proposed method might propose both O1 -> O2 and O2 -> O1, because they are close in the space of opinions, and the graph built via k nearest neighbors is undirected.


I wonder how unsupervised the method really is, in contrast with what the authors claim. It relies heavily on an opinion extractor, which rather means that when going from raw text to opinions, there is not really anything to learn, but any potential errors from this part of the pipeline are propagated to the factorization step.

A core component is the minimization of the reconstruction error of the item/opinion matrix, but the way this minimization is performed in practice is not specified.

These weaknesses are more related to how the work is conveyed, and I think the paper would benefit by discussing them. The actual impact of the proposed method and its relevance for the conference are still valuable.

---

> ### Author Response · Authors · 2020-04-08
> **Response to Reviewer #2 [1/1]**
>
> Please find our response to your questions/comments below.
>
> R2W1: Q: Is the opinion extractor nuanced?
> A: The opinion extractors can identify and incorporate negations, adverbial modifiers in the “modifier” for the target aspect. In the second example, the opinion extractor is expected to extract (unnecessarily complex, characters) as the opinion instead of (complex, characters). Both extractors we have used in our experiments capture these nuances and we assume other extractors useful as long as they can capture these subtleties.
>
> R2W2: Q: Does proximity between the opinion embeddings indicate implication?
> A: The proximity between the opinions embeddings, as the reviewer suggested, identifies similarity. However, once these similarities are computed, we proceed to create a K-nearest neighbor graph. A K-nearest neighbor graph is a directed graph and the direction of edges are what enable Sampo to discover implications. For instance, we see that “bad breakfast” is among the top 5 neighbors of “burnt coffee taste”, but “burnt coffee taste” does not appear in the top 5 neighbors of “bad breakfast” and more generic opinions such as “limited options”, “poor service” are closer to “bad breakfast”. This helps us identify that there should be a directed edge from “burnt coffee taste” to “bad breakfast” and not vice versa. On the other hand, “poor breakfast” and “bad breakfast” appear in the top 5 nearest neighbors of each other and we can conclude that the implication edge goes both ways which further implies the two opinions are equivalent. We will convey this more clearly in Section 3.3 in the uploaded revision.
>
> R2W3: Q: How unsupervised the method really is?
> A: Our problem formulation assumes access to an opinion extractor. Clearly, opinion extractors that are trained using supervised data tend to perform better. However, we pushed for relying on unsupervised (rule-based) extractors to ensure that the system can produce interesting results with no supervision. We experimented with rule-based extractors as well as supervised extractors. We found that while accuracy of the extractor can affect the quality of implications we find, it is possible to rely on highly-accurate pattern-based extractors (that might have limited recall) for domains where a labeled data for training an opinion extractor (that yields higher recall) is not available.
>
> R2W4: Q: What technique is used to minimize reconstruction error?
> A: As mentioned in the paper, we are using the package Tensorly to factorize the matrices. To factorize the matrix, Alternating Least Squares (ALS) technique to minimize the reconstruction error.

---

> > ### Comment · AnonReviewer1 · 2020-04-17
> > **thanks for clarification**
> >
> > Thanks for your clarifications, in particular on how you determine implication.
> >
> > Quick question: what do you mean by "pushed for relying on unsupervised extractors"?

---

> > > ### Author Response · Authors · 2020-04-17
> > > **response to clarification**
> > >
> > > We meant we used unsupervised extractors in our system to demonstrate that it can find interesting implications even by relying on simple rule-based extractors.

---

### Official Review · AnonReviewer3 · 2020-03-28
**Happy to see the unsupervised method, would like some clearer motivation**

**Rating:** 6
**Confidence:** 3

**Review:**

Summary of contributions:

This paper presents an entirely unsupervised method for taking as input a corpus of reviews about a set of items and building as output a directed graph where each node represents an opinion phrase (extracted from text) and edges indicate implication relationships.

After extracting opinion phrases from the review corpus (relying largely on prior work) the method constructs on a matrix where rows represent items and columns indicate opinions, with the cells containing counts of how often that opinion was expressed for the item. Matrix factorization is applied to produce embeddings for each opinion, with edges then created between k-nearest neighbors in the embedding space.

The contributions can be enumerated as:
- proposal of an entirely unsupervised system for construction of this opinion graph
- use of matrix factorization to determine similarity between opinions
- application of the method to data in several domains and analysis of results
- release of data and code


Strengths:

It is a fully unsupervised method that takes a corpus of text and produces a potentially useful piece of knowledge.

The matrix factorization approach is a simple but elegant way of discovering similarity between extracted phrases expressed in text.

Evaluations are conducted in several very different domains, from movies to electronics to restaurants, showing that the method is domain independent.

The comparisons with BERT and discussion of potential complement between the proposed approach and a language model approach is nice.

The paper is generally well written and easy to follow.


Weaknesses:

My primary complaint is that I would like to see more motivation of the utility of the constructed knowledge base. (I also somewhat disagree with the use of "knowledge base" to describe the outputed graph, as it contains a single relation and entity type, and that relation has ambiguous semantics.) Some specific issues that should be addressed:

- What is the use of such a graph? How would one use it in a downstream task?
- The abstract asserts that you show that your model can benefit downstream tasks such as review comprehension, but you do not directly show this. If you make this claim, you should provide an experiment showing improvement on this task by incorporating your graph.
- Why do we assume that proximity between the opinion embeddings indicates implication rather than just similarity? I assume that there will be some cases where two opinions in your graph imply each other, while in other cases there may be only a single edge (if one happens to have more nearby neighbors). Why should we consider these cases to be different?
- What exactly does implication mean in terms of opinions? I'd like to see a clearer definition here.

With regard to evaluation, did you measure the inter-annotator agreement between crowdworkers? The task of assessing "implication" between opinions seems quite muddy to me. In fact, I personally might disagree with your flagship example of "complex characters" implying "good writing". I don't think that is always true, and if I was given that example as a crowdworker I may have marked it as incorrect. Maybe my opinion would be in the minority, but this is why I would like to see a better and clearer motivation for the task.

With regard to your baselines, I was a little unclear on the application of universal schema. Universal schema uses a matrix where rows are entity pairs and columns are relations. You mention that you map all item-opinion pairs to a single "has-a" relation. Does that mean the matrix has only a single column?

The font size of Figures 2 and 3 is too small.

Minor point: There is a typo in the first full sentence on page 2, where "second review" should read "first review".

Overall Conclusion:

This paper presents a novel unsupervised method for extracting opinion phrases and implications between them from a corpus of reviews. The paper would benefit from better motivation for its problem and solution. That said, its unsupervised approach for knowledge extraction offers a useful method for capturing semantic relationships between textual phrases, and the experiments show strong performance against baselines.

---

> ### Author Response · Authors · 2020-04-08
> **Response to Reviewer #1 [1/2]**
>
> Please find our response to your questions/comments below.
>
> R1W1. Q: What is the motivation to build a KB with a single relation and entity type?
> A: It’s correct that our constructed KB contains a single relation. However, there are existing KBs in the literature that also contain a single relation. For example, Probase [1] only contains “isA” relation extracted from a text corpus. At the same time, we agree that humans (even when provided with a definition) might disagree over the semantics of the “implies” relation. However, this is quite common in KBs that deal with common sense. Most notable examples are ATOMIC and ConceptNet where they model relations such as “PersonX is seen as” and “X can be” respectively. These KB are still useful for downstream applications as many studies have shown.
> Similar to the examples mentioned above, we chose to refer to the output of our system as a knowledge base of opinions with implications as a relation which we automatically mine from a corpus of reviews.
>
> R1W2. Q: What is the use case of the KB?
> A: As mentioned in the paper, opinion KBs like ours are useful for downstream tasks such as reading comprehension tasks (e.g, question answering, sentiment analysis, etc.), retrieval and query expansion. For example, Google’s sentiment analysis tool (see cloud.google.com/natural-language) considers “The rooms in this hotel have really thin walls” as a neutral statement. Our constructed KB understands that “thin walls” implies “noisy room”. “Noisy room” has a negative sentiment according to Google’s API. Similarly, understanding the query “Show me a movie with good writing” requires the knowledge that “complex characters”, “fast paced action” imply good writing. Hence, instead of searching reviews with “good writing” as a keyword, a system could expand the search using our KB to additionally look for “complex characters” and “fast paced action”. We will convey this more clearly in the introduction of the revision.
>
> R1W3: Q: There are no experiments demonstrating improvements on RC task
> A: BERT is a state-of-the-art model for reading comprehension. Our experiments demonstrate that the implications we discovered are often missed by BERT-like models (as they fail to predict these implications) and that our KB can be used as training data to further tune these models and improve their performance on reading comprehension. We can make this more clear in our abstract.
>
> R1W4: Q: Does proximity between the opinion embeddings indicate implication?
> A: We have updated the paper to reflect this discussion better. The proximity between the opinions embeddings, as the reviewer suggested, identifies similarity. However, once these similarities are computed, we proceed to create a K-nearest neighbor graph. A K-nearest neighbor graph is a directed graph and the direction of edges are what enable Sampo to discover implications. Consider two opinions O1 and O2. There are many cases where O1 is among the top K neighbors of O2, but not vice versa. For instance, we see that “bad breakfast” is among the top 5 neighbors of “burnt coffee taste”, but “burnt coffee taste” does not appear in the top 5 neighbors of “bad breakfast” and more generic opinions such as “limited options”, “poor service” are closer to “bad breakfast”. This helps us identify that there should be a directed edge from “burnt coffee taste” to “bad breakfast” and not vice versa. On the other hand, “poor breakfast” and “bad breakfast” appear in the top 5 nearest neighbors of each other and we can conclude that the implication edge goes both ways which further implies the two opinions are equivalent. We will write this more clearly in Section 3.3 in the next revision.
>
> R1W5: Q: Definition of implication?
> A: Here is the definition that we provided to the crowdworkers. Given two opinions O1 and O2, we consider O1 → O2, if opinion O1 implies or contributes to O2. By “implies”, we mean that O1 is an expression that essentially conveys O2 (e.g., thin walls → noisy room). By “contributes”, we mean that O1 is a valid reason for O2 to be true. For instance, “thin walls” → “bad hotel” falls into this category as having “thin walls” contributes to a hotel being bad but it may not be sufficient to consider the hotel as necessarily bad.
>
> R1W6: Q: What is the inter-annotator agreement?
> A: There are certainly cases where human annotators don’t agree with each other, but in general we observed a good degree of agreement among the annotators. While we have collected a single judgement for almost all mined relationships, we set aside a set of 70 questions (i.e, 10-15 questions per domain) to which all crowdworkers (roughly 1200 participants) must have provided an answer. We have observed a Fleiss Kappa agreement score of 0.76 when considering the following 4 categories: “O1 implies O2”, “O2 implies O1”, “O1 and O2 are equivalent”, “O1 and O2 are unrelated”. We have updated the appendix with the agreement scores to reflect this.

---

> > ### Author Response · Authors · 2020-04-08
> > **Response to Reviewer #1 [2/2]**
> >
> > Please find our response to your questions/comments below.
> >
> > R1W7: Q: Details of the universal Schema experimental set-up?
> > A: We basically considered the items being reviewed as entities and considered each opinion O as a unary relation which we called “has-a-O”. For instance, we consider “Casablanca” has-a-(romantic,theme). Thus, our matrix has as many relations as our extracted opinions. Universal schema computes a representation for each relation and we use these representations to measure the similarity of between different opinions. We have updated section 4.1 to convey this clearly.
> >
> > R1W8: Q: Redo figures
> > A: These figures are updated in the uploaded revision.
> >
> > R1W9: Q: Typos
> > A: We have modified the text to better reflect our point. In fact, the relation specified in the example is implied by the fact that both reviews describe the same movie.

---

### Author Response · Authors · 2020-04-08
**Thank you to our reviewers!**

We thank the reviewers for their encouraging and detailed feedback. We have uploaded the revision to include more details and clarifications.

---

### Decision · Program_Chairs · 2020-05-01

**Decision:**

Accept

**Comment:**

This paper addresses the task of unsupervised knowledge base construction. The reviewers like that the authors present a novel unsupervised approach, and are happy with the thorough experiments. However, they also point out that the approach could be motivated better, and that it makes many assumptions that are not explained properly.  We recommend acceptance but nudge the authors to consider the reviewer suggestions.